# Force-velocity profiling in athletes: Reliability and agreement across methods

Kolbjørn Lindberg[1,2]*, Paul Solberg[2], Thomas Bjørnsen[1,2], Christian Helland[2], Bent Rønnestad[2,3], Martin Thorsen Frank[1], Thomas Haugen[2,4], Sindre Østerås[2,5], Morten Kristoffersen[2,6], Magnus Midttun[2], Fredrik Sæland[2], Gøran Paulsen[2,7]

1 Department of Sports Science and Physical Education, Faculty of Health and Sport Sciences, University of Agder, Kristiansand, Norway, 2 Norwegian Olympic and Paralympic Committee and Confederation of Sports, Oslo, Norway, 3 Department of Health and Exercise Physiology, Faculty of Social Sciences, Inland Norway University of Applied Sciences, Lillehammer, Norway, 4 School of Health Sciences, Kristiania University College, Oslo, Norway, 5 Department of Neuromedicine and Movement Science, Faculty of Medicine and Health Sciences, Centre for Elite Sports Research, Norwegian University of Science and Technology, Trondheim, Norway, 6 Department of Sport and Education, Bergen University College, Bergen, Norway, 7 Department of Physical Performance, Norwegian School of Sport Sciences, Oslo, Norway

* kolbjorn.a.lindberg@uia.no

**Data Availability Statement:** All relevant data are within the manuscript and its Supporting Information files.

## Abstract

The aim of the study was to examine the test-retest reliability and agreement across methods for assessing individual force-velocity (FV) profiles of the lower limbs in athletes. Using a multicenter approach, 27 male athletes completed all measurements for the main analysis, with up to 82 male and female athletes on some measurements. The athletes were tested twice before and twice after a 2- to 6-month period of regular training and sport participation. The double testing sessions were separated by ~1 week. Individual FV-profiles were acquired from incremental loading protocols in squat jump (SJ), countermovement jump (CMJ) and leg press. A force plate, linear encoder and a flight time calculation method were used for measuring force and velocity during SJ and CMJ. A linear regression was fitted to the average force and velocity values for each individual test to extrapolate the FV-variables: theoretical maximal force ($F_0$), velocity ($V_0$), power ($P_{max}$), and the slope of the FV-profile ($S_{FV}$). Despite strong linearity ($R^2 > 0.95$) for individual FV-profiles, the $S_{FV}$ was unreliable for all measurement methods assessed during vertical jumping (coefficient of variation (CV): 14–30%, interclass correlation coefficient (ICC): 0.36–0.79). Only the leg press exercise, of the four FV-variables, showed acceptable reliability (CV:3.7–8.3%, ICC:0.82–0.98). The agreement across methods for $F_0$ and $P_{max}$ ranged from (Pearson r): 0.56–0.95, standard error of estimate (SEE%): 5.8–18.8, and for $V_0$ and $S_{FV}$ r: -0.39–0.78, SEE%: 12.2–37.2. With a typical error of 1.5 cm (5–10% CV) in jump height, $S_{FV}$ and $V_0$ cannot be accurately obtained, regardless of the measurement method, using a loading range corresponding to 40–70% of $F_0$. Efforts should be made to either reduce the variation in jumping performance or to assess loads closer to the FV-intercepts. Coaches and researchers should be aware of the poor reliability of the FV-variables obtained from vertical jumping, and of the differences across measurement methods.

**Funding:** The author(s) received no specific funding for this work.

**Competing interests:** The authors have declared that no competing interests exist.

## Introduction

Within strength and power training, force-velocity (FV) profiling has received increasing attention as a means to monitor training adaptations [1–3] and to serve as a basis for individual training prescriptions for athletes [3–6]. The concept of FV-profiling is based on the fundamental properties of skeletal muscles, where there is an inverse relationship between force and velocity [7].

In multi-joint movements, the FV-relationship is commonly described as linear [8], in contrast to the hyperbolic relationship observed in isolated muscles or single-joint movements [7]. In practice, athletes can perform maximal efforts against different loads while force and velocity are measured during vertical jumping or similar multi-joint movements. Based on such data, one can draw a linear regression line and extrapolate the theoretical maximal force ($F_0$) (i.e., force at zero velocity) and velocity ($V_0$) (i.e., velocity at zero force). Following that, the theoretical maximal power ($P_{max}$) can be calculated as ($F_0 \cdot V_0$)/4 and the slope of the FV-profile ($S_{FV}$) as $F_0/V_0$ [9]. However, controversy exists about the linearity of FV-relationships obtained from multi-joint movements [8].

The value of a test is highly dependent on its reliability, especially when evaluating individual data from high-performing athletes [10]. However, although several studies have evaluated the within-session reliability of FV-variables [11–18], limited attention has been directed towards the between-session reliability of these FV-variables in athletes. Additionally, only encoders and the flight time calculation method have been used for measurements of between-session reliability of the FV-variables [12, 13, 19]. Hence, the reliability of other commonly used methods such as force plates and leg press devices is unknown [11–18]. Furthermore, different devices and methods (e.g., force plates, linear position transducers, pneumatic resistance apparatus and the flight time calculation method) are used to assess the lower limb FV-variables, but the agreement among these has received limited attention [17, 20–22].

Giroux et al. [20] previously investigated the reliability and agreement among three measurement methods (accelerometry, linear position transducer and flight time calculation method) during vertical jumps. However, they reported only average values of force, velocity and power for each jump, and not the extrapolated FV-parameters ($F_0$, $V_0$, $P_{max}$ and $S_{FV}$) that are increasingly used for individual training prescriptions [3–5, 23]. García-Ramos et al. [22] investigated the agreement across methods for CMJ (force platform, linear position transducer and flight time calculation method), but not SJ. As the test-retest reliability of the different methods for assessing individual FV-profiles is of crucial importance, it is of great interest to investigate the mentioned shortcomings in the literature.

A novel aspect of FV-profiling during vertical jumping is the possibility of obtaining the extrapolated variable $V_0$ and the calculated $S_{FV}$, as there are numerous methods for assessing maximal force and maximal power [24]. Interestingly, $S_{FV}$ and $V_0$ have previously shown poorer reliability than $F_0$ and $P_{max}$ in vertical jumping [11]. Cuk et al. [25] hypothesized that this lower reliability might be due to the distance of extrapolation, as all measurements are performed closer to $F_0$ compared to $V_0$, in addition to the small range in loads assessed during incremental loading protocols in vertical jumping. These speculations were partly confirmed by García-Ramos et al. [26], who reported that the load range used to acquire the FV-profile significantly affects the reliability of $V_0$. Assessing loads close to $F_0$ is limited by the technical demand of jumping with heavy loads, while attempts closer to $V_0$ are limited by the subject's own bodyweight during vertical jumping. However, the bodyweight issue is not present during the leg press exercise, making it possible to assess loads closer to both $F_0$ and $V_0$, potentially improving the reliability for the FV-variables. It is therefore of great interest to investigate the

reliability of the extrapolated FV-variables from commonly used vertical jumping exercises as well as from the leg press exercise.

The aim of the present study was to examine the i) test-retest reliability and ii) agreement across methods for assessing individual FV-profiles of the lower limbs in well-trained athletes.

## Methods

### Experimental approach and design

The participants in the present study underwent physical testing four times. The first two testing timepoints were separated by ~1 week, before a training period of 2~6 months. The two last timepoints were also separated by ~1 week (Figs 1 and 2).

The data were collected from multiple regional Olympic training and testing centers. Because not all facilities had the same testing capacities, the sample size differed across the measurement methods. Therefore, the main analysis in this study was performed on the participants tested under all methods (reliability and agreement), with an additional aggregated analysis including all participants, with varying sample sizes across methods (only reliability analysis). For the main analysis, the test leaders were constant, and for the aggregated analysis the test leaders and equipment differed across centers but were kept constant for each participant (sample sizes for all tests are presented in the results section). Written informed consent was obtained from all participants prior to commencing their involvement in the study.

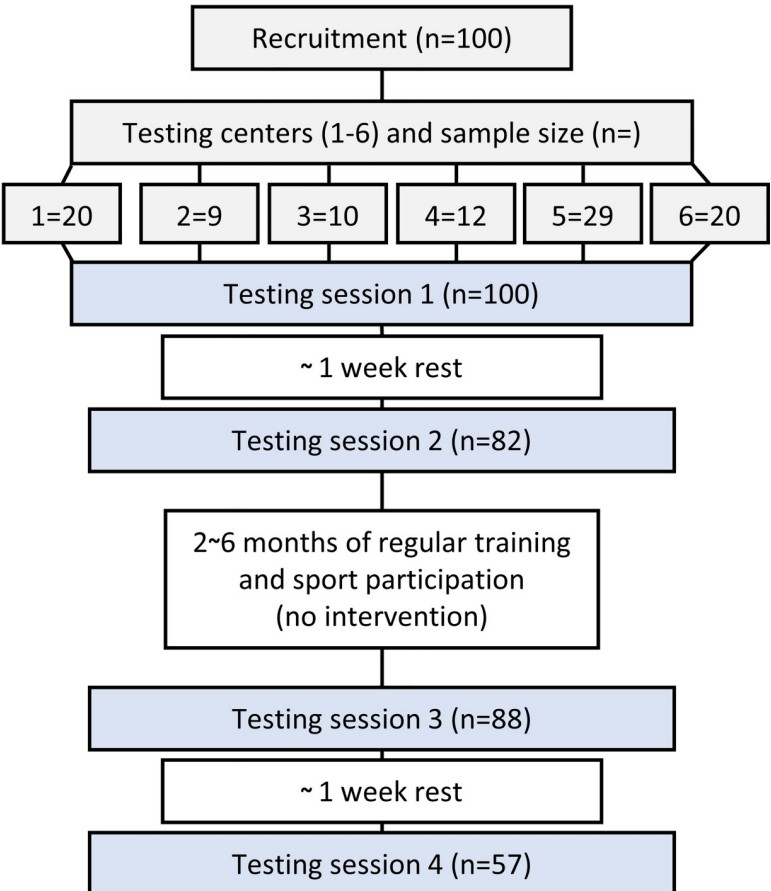

**Fig 1. Flow chart representing study design.**

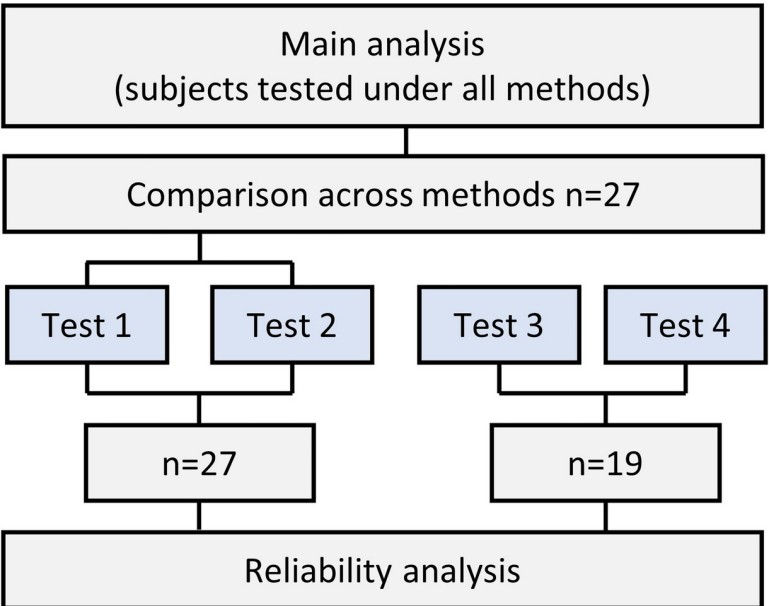

**Fig 2. Flow chart representing study design and sample size for main analysis.**

The study was reviewed by the ethical committee of Inland Norway University of Applied Sciences, approved by the Norwegian Centre for Research Data and performed in agreement with the Declaration of Helsinki. The athletes in the main sample were familiar with the testing procedures, whereas the subjects in the mixed sample had various levels of experience prior to the study.

**Participants.**   For the main analysis, a total of 27 well-trained male athletes from handball and ice hockey were included (age 21 ± 5 years; height 185 ± 8 cm; body mass 84 ± 13 kg; Table 1).

For the aggregated mixed sample, both male (approximately 80% of sample) and female athletes participated (age 21 ± 4 years; height 182 ± 9 cm; body mass 78 ± 12 kg; Table 2). Most of the participants were team sport players in handball, ice hockey, soccer, and volleyball, while the remaining participants competed in Nordic combined, ski jumping, weightlifting, athletics, badminton and speed skating. The competition level ranged from world class (Olympic medalist) to club level, with the majority competing at national and international level in their respective sports.

**Testing procedures.**   All participants were instructed to prepare for the test days as they would for a regular competition in terms of nutrition, hydration, and sleep, and to refrain from strenuous exercise 48 hours prior to testing. All testing was performed indoors, and the participants were instructed to use identical footwear and clothing on each test day.

Bodyweight was measured wearing training clothes and shoes (as total bodyweight is used to calculate force in some of the methods). All participants performed a standardized ~10-min

**Table 1.  Performance characteristics of the athletes for main analysis.**

|  | *Mean ± SD* | *Max* | *Min* |
|---|---|---|---|
| *CMJ (cm)* | 38 ± 4 | 43 | 28 |
| *SJ (cm)* | 36 ± 4 | 43 | 28 |

Values from baseline measures, sample size = 27, SJ: Squat jump, CMJ: Countermovement jump, cm: Centimeters, s: seconds, SD: Standard deviation.

**Table 2. Performance characteristics of the athletes for aggregated analysis.**

|  | n = | Mean ± SD | Max | Min |
|---|---|---|---|---|
| CMJ (cm) | 83 | 38 ± 5 | 58 | 25 |
| SJ (cm) | 72 | 35 ± 6 | 51 | 22 |

Values from baseline measures, sample size in table. SJ: Squat jump, CMJ: Countermovement jump, Centimeters, s: seconds, SD: Standard deviation.

warm-up procedure prior to testing, consisting of jogging, local muscle warm-up (hamstring and hip mobility–consisting of light dynamic stretches), running drills (e.g. high knees, skipping, butt-kicks, explosive lunges) and bodyweight jumps.

The different tests were separated by 5–10 min to ensure proper recovery, and light snacks and drinks were offered to the participants during the testing sessions. The testing protocol consisted of a series of squat jumps (SJ), countermovement jumps (CMJ) and a leg press test with incremental loads.

SJ and CMJ were initially performed with bodyweight, accompanied by an incremental loading protocol consisting of 0.1 (broomstick), 20, 40, 60 and 80 kg. In the aggregated sample, for some weaker participants (i.e., those unable to jump with 80 kg), a protocol of approximately 5 loads up to 80% of bodyweight was used. The increase in loads was then individually determined. In both the SJ and CMJ, the FV-relationship was derived from a force plate (For main analysis: Musclelab; Ergotest AS, Porsgrunn, Norway and for aggregated analysis some tested at: AMTI; Advanced Mechanical Technology, Inc Waltham Street, Watertown, USA) and a linear position transducer encoder (Ergotest AS, Porsgrunn, Norway). The encoder was placed on the ground and connected to the barbell. Participants were instructed to keep their hands on their hips for the bodyweight trials, and a broomstick was used as the 0.1 kg load. Two valid trials were registered for each load. The recovery after each attempt was 2–3 min.

For the SJ, participants were asked to maintain their individual starting position (∼90˚ knee angle) for about 2 s and then apply force as fast as possible and jump to the maximum possible height before landing with their ankles in an extended position. Countermovement was not allowed for the SJ and was checked visually with the direct force output from the force plate. The starting position for both SJ and CMJ was standardized to the athlete's self-selected starting position and kept constant for all jumps and testing sessions. The starting position for the SJ and the depth of the CMJ was controlled using a rubber band beneath the thighs of the athletes. If these requirements were not met, the trial was repeated. The CMJ test procedure was similar to that for SJ, except for the pause in the bottom position.

For the leg press, Keiser Air300 horizontal pneumatic leg press equipment with an A420 force and velocity measuring device (Keiser Sport, Fresno, CA) was used. The FV-variables were derived from a 10-repetition FV-test pre-programmed in the Keiser A420 software. To determine the loading range, each participant's 1RM was obtained at the familiarisation session for the main analysis, whereas the 1RM was individually estimated for the participants in the aggregated analysis. The test started with two practice attempts at the lightest load, corresponding to ∼15% of 1RM. Thereafter, the load was gradually increased with fixed steps (∼20–30 kgf) for each attempt until reaching the ∼1RM load and a total of 10 attempts across the FV-curve (15–100% of 1RM). The rest period between attempts got longer as the load increased. The rest period between attempts was ∼10–20 seconds for the initial five loads, and 20–40 seconds for the last four loads. The seating position was adjusted for each participant, aiming at a vertical femur, equivalent to an 80-90˚ knee angle, and the feet were placed with the heels at the lower end of the foot pedal. Participants were asked to extend both legs using

maximum effort during the entire 10-repetition FV-test. Due to the pneumatic semi-isotonic resistance, maximal effort does not cause ballistic action, and the entire push-off was performed with maximal intentional velocity. The leg press was performed as a concentric-only action without countermovement, as the pedals were resting in their predetermined position prior to each repetition. The eccentric phase was submaximal and not registered.

### Data analysis

All FV-variables were obtained from the average force and velocity during the concentric phase of the movement. For each incremental loading test, a linear regression was fitted to the average force and velocity measurements to calculate the individual FV-variables. $F_0$ and $V_0$ were defined as the intercepts of the linear regression for the corresponding force and velocity axis, while $S_{FV}$ refers to the slope of the linear regression. $P_{max}$ was then calculated as $F_0 \cdot V_0/4$. All FV-variables were obtained from FV-profiles with a coefficient of determination greater than 0.95 [9].

Force plate: FV-variables derived from the force plate were analysed using a customized Microsoft Excel spreadsheet (Microsoft Office Professional Plus 2018, version 16.23). Velocity was calculated by integrating the acceleration obtained from the ground reaction forces. The centre of mass position was the integral of velocity, while power was the product of force and velocity [27]. The start of the concentric phase for the SJ was defined as the point at which force exceeded 5 SD of the steady-stance weight prior to the jump [27–29]. For the CMJ, the integration of velocity started when the force fell below 5 SD of the steady-stance weight. The concentric phase was defined as the point at which velocity was greater than 0 m/s. The end of the concentric phase for both SJ and CMJ was defined as the instant when the participant left the force plate (i.e., take-off: when forces fell below 10N).

Encoder: By measuring the position of the cable (connected to the bar) as a function of time, the software calculates force and velocity (MuscleLab, version 10.5.69.4815). Average force was calculated as the product of mass and acceleration. Acceleration was calculated as the average velocity divided by the duration of the positive displacement, with the addition of the gravitation constant, while mass was calculated as bodyweight plus external load. In agreement with the manufacturer´s recommendation and previous studies [30], 90% of body mass and 100% of external load were used to calculate force during SJ and CMJ. *Flight time method*: Average force ($\bar{F}$) and average velocity ($\bar{v}$) were calculated using two equations, considering only simple input variables: body mass, jump height and push-off distance [15, 31]. The vertical push-off distance was determined as previously proposed [9], corresponding to the difference between the extended lower limb length with maximal foot plantar flexion and the crouch starting position of the jump.

Keiser leg press: The Keiser Air300 horizontal leg press dynamometer uses pneumatic resistance and measures compression forces at the cylinder, while velocity is measured with a position transducer. The values at the cylinder are then calculated to match the range of motion and velocity at the apparatus pedals [1]. Average force and velocity were calculated as a function of time, where the software excludes 5% of the range of motion from the start and end of the movement.

The measurement sample rate for the MuscleLab force plate and encoder was 200 Hz and for the leg press apparatus was 400 Hz. The force signal from the Musclelab force plate data was upsampled to 1000 Hz by spline integration using the integrated software. The AMTI force plate sampled at 2000 Hz.

### Statistical analysis

The coefficient of variation (CV%), interclass correlation coefficient (ICC 3,1) and mean percent change (%Δ) were used to assess reliability across the testing sessions. CV% and %Δ were

calculated from the log-transformed data. The Pearson product-moment correlation coefficient (Pearson r) was used to determine the association across methods. For comparison across methods, the mean difference (systematic bias) was calculated and presented in absolute and in relative terms (% from log transformed data) with percent and standardized difference (mean difference divided by the standard deviation of the criterion measure).

The standardized difference was qualitatively interpreted using the scale (<0.2 Trivial; 0.2–0.6 Small; 0.6–1.2 Moderate; 1.2–2.0 Large; 2.0–4.0 Very large; >4.0 Extremely large) [32]. A paired sample t-test was used to test the significance level of the differences in means. Additionally, a linear regression analysis with corresponding slope and Y-intercept of the regression line was used for comparison across methods. The standard error of the estimate (SEE) was calculated from the linear regression and presented in absolute and relative terms. For comparison across methods, the averages of the two first testing timepoints were included.

The smallest worthwhile change (SWC%) was calculated as 0.2 of the between-athlete SD, presented as a percentage of the mean. Confidence limits (CL) for all analyses were set at 95%. The Pearson's r coefficients were interpreted categorically (<0.09 trivial; 0.10–0.29 small; 0.30–0.49 moderate; 0.50–0.69 large; 0.70–0.89 very large; 0.90–0.99 nearly perfect; 1.00 perfect) as defined by Hopkins and Marshall [33].

Acceptable reliability was considered as ICC $\geq$ 0.80 and CV $\leq$ 10%, while good reliability was considered as ICC $\geq$ 0.90 and CV $\leq$ 5% [34–41]. Descriptive data are reported as mean ± SD. All statistical analyses were performed using a customized Microsoft Excel spreadsheet [32].

## Results

### Test-retest reliability of the FV-variables

All FV-profiles displayed linearity, with individual $R^2$ values ranging from 0.95 to 1.00. All the following results presented in the text correspond to results from the main analysis, whereas results from the aggregated analysis are only presented in tables. Fig 3 and Table 3 show the reliability measures of the FV-variables for the main analysis. Table 4 shows the reliability measures of the FV variables for the aggregated analysis.

Of all the investigated measurement methods, only the leg press showed acceptable reliability for the four FV-variables (CV: 3.7–8.3%, ICC: 0.82–0.98). Several of the measures for $P_{max}$ and $F_0$ obtained from the vertical jumps showed acceptable reliability (CV: 3.9–12.1%, ICC: 0.61–0.97) (Table 3). However, $V_0$ and $S_{FV}$ showed unacceptable reliability for all the investigated SJ and CMJ measurement methods (CV: 8.4–30.1%, ICC: 0.16–0.79). The typical error for both SJ and CMJ jump height was 1.2 cm, corresponding to a coefficient of variation of 6.8%. For each loading condition (0, 20, 40, 60 and 80 kg) the typical error was: 1.7, 1.2, 0.9, 1.0 and 1.0 cm corresponding to a CV of 5.1, 4.6, 5.5, 7.6 and 10.2% respectively.

### Agreement across methods

The agreement and comparisons for the different measurement methods are shown in Table 5. Mean±SD values for all the FV-methods are shown in Table 6 and illustrated in Fig 4. The agreement across methods for $F_0$ and $P_{max}$ ranged from (Pearson r): 0.56–0.95, SEE%: 5.8–18.8, and for $V_0$ and $S_{FV}$ r: -0.39–0.78, SEE%: 12.2–37.2. The mean bias for $F_0$ ranged from trivial to moderate (-6-14%, ES: -0.4–0.9); small to large for $P_{max}$ (-30-55%, ES: -1.8–1.7); trivial to very large for $V_0$ (-35-70%, ES: -2.8–2.2); and small to very large for $S_{FV}$ (-32-165%, ES: -1.2–3.8) (Tables 5 and 6 and Fig 4).

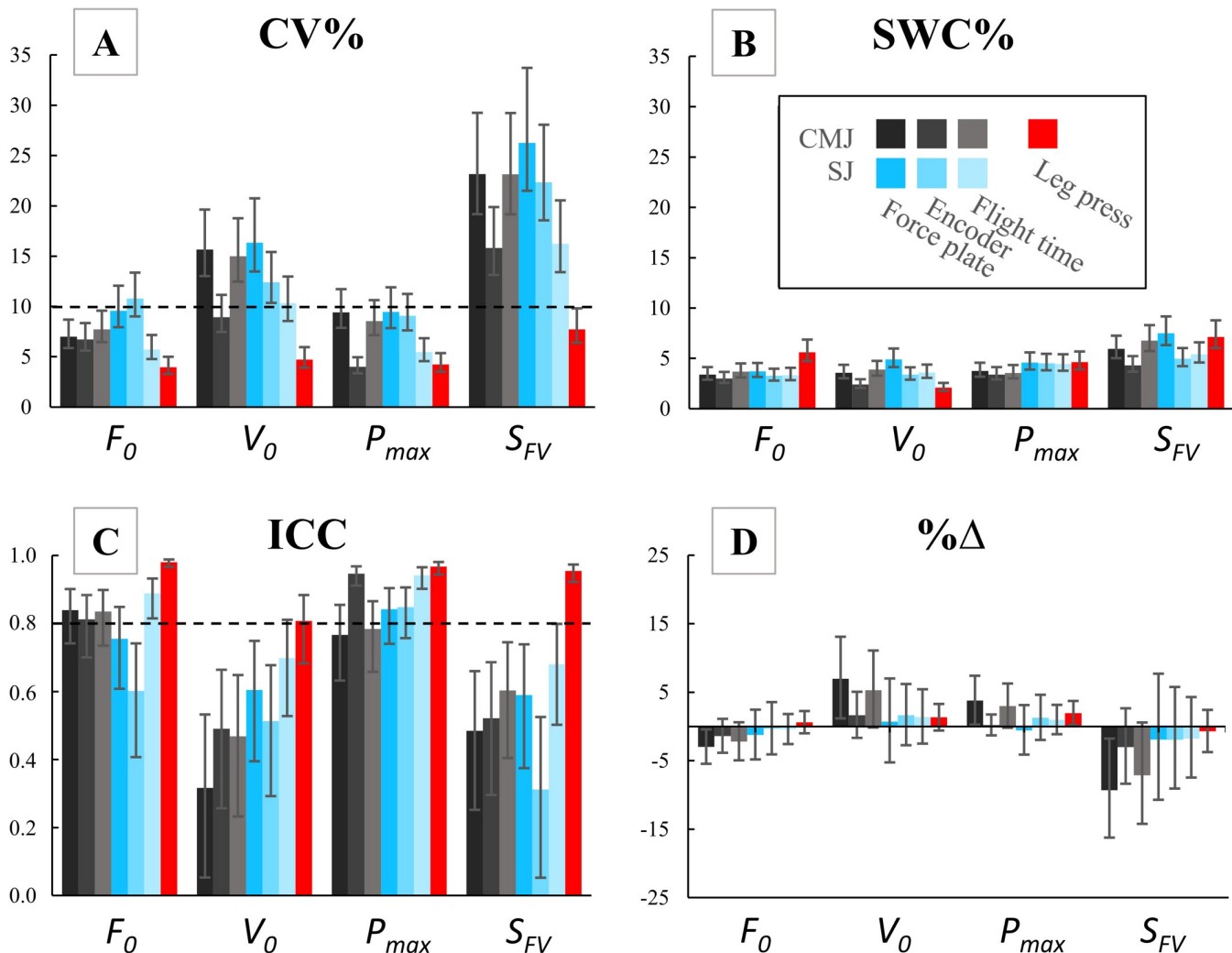

**Fig 3. Measures of reliability for the FV variables obtained from main analysis.** Panel A- Coefficient of variation (CV%), panel B- Smallest worthwhile change (SWC%), panel C- Interclass correlation coefficient (ICC), panel D- Mean % change (%Δ). All values were obtained by combining test 1–2 (n = 27) and 3–4 (n = 19). Error bars represent 95% confidence intervals. Dotted line represents line of acceptable reliability.

## Discussion

This is the first study to investigate the between-session reliability of FV-profiles measured in SJ and CMJ with a force plate, linear encoder, and a flight time calculation method, in addition to a leg press task. The main finding of the present study was that regardless of strong linearity for individual FV-profiles, $S_{FV}$ and $V_0$ were unreliable for all measurement methods assessed from vertical jumping using loads ranging from bodyweight to 80 kg (relative position on the FV-curve, force values 40–70% of $F_0$). Only the leg press exercise showed acceptable reliability for the four FV-variables (relative position on the FV-curve, force values 20–80% of $F_0$). There was a large to nearly perfect association across measurement methods for $F_0$ and $P_{max}$, while the association for $V_0$ and $S_{FV}$ ranged from trivial to large.

### Test-retest reliability of the FV-variables

To the authors' knowledge, this is the first study to assess the test-retest reliability of the FV-variables in well trained and elite athletes. The present results are in accordance with previous

**Table 3. Measures of reliability for the FV variables obtained from the main analysis with corresponding 95% confidence intervals.**

| | | Coefficient of variation (CV%) | | | | Interclass correlation (ICC) | | | | Percent change (%Δ) | | | |
|---|---|---|---|---|---|---|---|---|---|---|---|---|---|
| | Test | $F_0$ | $V_0$ | $P_{max}$ | $S_{FV}$ | $F_0$ | $V_0$ | $P_{max}$ | $S_{FV}$ | $F_0$ | $V_0$ | $P_{max}$ | $S_{FV}$ |
| CMJ Force plate | 1–2 | **8.6 ± 2.6** | 19.2 ± 6.2 | 10.8 ± 3.4 | 29.0 ± 9.8 | **0.81 ± 0.14** | 0.20 ± 0.37 | 0.74 ± 0.18 | 0.40 ± 0.32 | -2.3 ± 4.5 | 6.5 ± 10.5 | 4.0 ± 6 | -8.3 ± 13.1 |
| | 3–4 | **5.1 ± 1.8** | 12.6 ± 4.6 | 8.8 ± 3.1 | 17.5 ± 6.5 | **0.89 ± 0.10** | 0.16 ± 0.43 | 0.77 ± 0.19 | 0.47 ± 0.34 | -2.6 ± 3.1 | 7.1 ± 8.2 | 4.3 ± 5.7 | -9.1 ± 9.5 |
| CMJ Encoder | 1–2 | **6.8 ± 2** | 9.8 ± 2.9 | **4.4 ± 1.3** | 16.9 ± 5.2 | **0.82 ± 0.13** | 0.43 ± 0.30 | **0.95 ± 0.04** | 0.47 ± 0.29 | -3.1 ± 3.4 | 3.9 ± 5.2 | 0.6 ± 2.3 | -6.7 ± 7.8 |
| | 3–4 | 7.0 ± 2.5 | 8.4 ± 3.1 | **3.9 ± 1.4** | 15.5 ± 5.9 | 0.78 ± 0.19 | 0.44 ± 0.37 | **0.95 ± 0.04** | 0.38 ± 0.39 | 1.4 ± 4.5 | -1.8 ± 5.3 | -0.4 ± 2.5 | 3.2 ± 9.9 |
| CMJ Flight time | 1–2 | 10.1 ± 3.1 | 18.7 ± 6 | 9.6 ± 2.9 | 30.1 ± 10.2 | 0.79 ± 0.15 | 0.29 ± 0.35 | 0.74 ± 0.18 | 0.50 ± 0.29 | -3.0 ± 5.2 | 4.4 ± 10 | 1.2 ± 5.2 | -7.1 ± 13.7 |
| | 3–4 | **5.2 ± 1.8** | 11.8 ± 4.3 | **7.8 ± 2.8** | 16.9 ± 6.3 | **0.92 ± 0.08** | 0.70 ± 0.23 | 0.82 ± 0.15 | 0.79 ± 0.18 | -1.7 ± 3.2 | 7.7 ± 7.8 | 5.9 ± 5.1 | -8.8 ± 9.2 |
| SJ Force plate | 1–2 | 11.2 ± 3.5 | 17.4 ± 5.6 | **9.4 ± 2.9** | 29.3 ± 9.9 | 0.69 ± 0.21 | 0.60 ± 0.25 | **0.87 ± 0.10** | 0.51 ± 0.29 | 0.5 ± 6.0 | -2.7 ± 8.8 | -2.2 ± 4.9 | 3.2 ± 14.9 |
| | 3–4 | **6.7 ± 2.4** | 15.4 ± 5.7 | 10 ± 3.6 | 22.3 ± 8.5 | **0.84 ± 0.13** | 0.54 ± 0.32 | 0.81 ± 0.16 | 0.57 ± 0.30 | -2.2 ± 4.1 | 4.1 ± 9.6 | 1.8 ± 6.2 | -6.0 ± 12.2 |
| SJ Encoder | 1–2 | 12.1 ± 3.5 | 11.1 ± 3.2 | 11.5 ± 3.4 | 21.0 ± 6.4 | 0.61 ± 0.24 | 0.59 ± 0.24 | 0.81 ± 0.13 | 0.36 ± 0.32 | 2.0 ± 6.1 | -1.4 ± 5.5 | 0.6 ± 5.8 | 3.4 ± 10.4 |
| | 3–4 | 6.5 ± 2.2 | 10.2 ± 3.6 | **5.2 ± 1.8** | 16.9 ± 6.1 | 0.77 ± 0.18 | 0.62 ± 0.27 | **0.94 ± 0.05** | 0.42 ± 0.36 | -3.0 ± 3.8 | 6.0 ± 6.5 | 2.9 ± 3.3 | -8.5 ± 9.0 |
| SJ Flight time | 1–2 | **5.2 ± 1.6** | 8.6 ± 2.6 | **4.4 ± 1.3** | 13.9 ± 4.4 | **0.92 ± 0.06** | 0.79 ± 0.16 | **0.97 ± 0.03** | 0.76 ± 0.17 | 0.8 ± 2.9 | -2.7 ± 4.5 | -1.9 ± 2.4 | 3.7 ± 7.5 |
| | 3–4 | **6.4 ± 2.3** | 11.6 ± 4.2 | **5.8 ± 2** | 18.5 ± 7.0 | **0.86 ± 0.13** | 0.63 ± 0.27 | **0.93 ± 0.07** | 0.62 ± 0.28 | -1.7 ± 3.9 | 6.7 ± 7.6 | 4.9 ± 3.8 | -7.9 ± 10.1 |
| Keiser leg press | 1–2 | **4.2 ± 1.3** | **5.0 ± 1.5** | **4.2 ± 1.3** | **8.3 ± 2.5** | **0.98 ± 0.02** | **0.82 ± 0.14** | **0.97 ± 0.02** | **0.95 ± 0.04** | 0.2 ± 2.3 | 2.2 ± 2.8 | 2.4 ± 2.4 | -2.0 ± 4.4 |
| | 3–4 | **3.7 ± 1.4** | **4.3 ± 1.6** | **4.2 ± 1.6** | **7.0 ± 2.6** | **0.98 ± 0.02** | **0.82 ± 0.16** | **0.97 ± 0.03** | **0.96 ± 0.04** | 1.3 ± 2.5 | 0.4 ± 2.9 | 1.7 ± 2.9 | 0.9 ± 4.6 |

Bold text denotes CV<10% and ICC>0.80. Sample size for test 1–2 = 27, and test 3–4 = 19. SJ: Squat jump, CMJ: Countermovement jump, $F_0$:Theoretical maximal force, $V_0$: Theoretical maximal velocity, $P_{max}$: Theoretical maximal power, $S_{FV}$: slope of the force-velocity profile.

research in other populations showing mostly acceptable reliability for $F_0$ and $P_{max}$ (CV<10%) and poor reliability for $V_0$ and $S_{FV}$ (CV >10%) during vertical jumping [12, 19, 25, 42, 43]. In contrast, FV-profiles derived from the leg press exercise displayed acceptable reliability for all variables in the present study (CV<10%, ICC>0.8). Feeney et al. [11] proposed that the low reliability

**Table 4. Measures of reliability for the FV variables obtained from aggregated analysis with corresponding 95% confidence intervals.**

| | | | Coefficient of variation (CV%) | | | | Interclass correlation (ICC) | | | | Percent change (%Δ) | | | |
|---|---|---|---|---|---|---|---|---|---|---|---|---|---|---|
| | Test | n = | $F_0$ | $V_0$ | $P_{max}$ | $S_{FV}$ | $F_0$ | $V_0$ | $P_{max}$ | $S_{FV}$ | $F_0$ | $V_0$ | $P_{max}$ | $S_{FV}$ |
| CMJ Force plate | 1–2 | 34 | **8.0 ± 2.1** | 17.5 ± 4.9 | 9.9 ± 2.7 | 26.5 ± 7.7 | **0.81 ± 0.12** | 0.22 ± 0.32 | 0.76 ± 0.15 | 0.40 ± 0.29 | -3.2 ± 3.7 | 6.9 ± 8.5 | 3.4 ± 4.8 | -9.4 ± 10.5 |
| | 3–4 | 21 | **5.1 ± 1.8** | 12.6 ± 4.6 | 8.8 ± 3.1 | 17.5 ± 6.5 | **0.89 ± 0.10** | 0.19 ± 0.43 | 0.78 ± 0.18 | 0.45 ± 0.35 | -2.6 ± 3.1 | 7.1 ± 8.2 | 4.3 ± 5.7 | -9.1 ± 9.5 |
| CMJ Encoder | 1–2 | 82 | **6.8 ± 1.1** | 8.6 ± 1.4 | **4.0 ± 0.6** | 15.5 ± 2.6 | **0.89 ± 0.05** | 0.74 ± 0.10 | **0.96 ± 0.02** | 0.78 ± 0.09 | -2.4 ± 2.0 | 2.2 ± 2.6 | -0.3 ± 1.2 | -4.5 ± 4.3 |
| | 3–4 | 56 | 7.3 ± 1.5 | 9.4 ± 1.9 | **3.7 ± 0.7** | 17.0 ± 3.6 | **0.81 ± 0.09** | 0.51 ± 0.19 | **0.96 ± 0.02** | 0.48 ± 0.20 | -0.7 ± 2.6 | 0.5 ± 3.4 | -0.2 ± 1.4 | -1.1 ± 5.9 |
| CMJ Flight time | 1–2 | 34 | **9.0 ± 2.4** | 16.8 ± 4.7 | 8.8 ± 2.4 | 26.7 ± 7.8 | **0.80 ± 0.13** | 0.31 ± 0.31 | 0.78 ± 0.14 | 0.51 ± 0.26 | -2.5 ± 4.2 | 3.8 ± 8.0 | 1.2 ± 4.2 | -6.1 ± 11 |
| | 3–4 | 21 | **5.2 ± 1.8** | 11.8 ± 4.3 | **7.8 ± 2.8** | 16.9 ± 6.3 | **0.92 ± 0.08** | 0.69 ± 0.24 | 0.81 ± 0.16 | 0.78 ± 0.18 | -1.7 ± 3.2 | 7.7 ± 7.8 | 5.9 ± 5.1 | -8.8 ± 9.2 |
| SJ Force plate | 1–2 | 45 | 10.8 ± 2.5 | 15.3 ± 3.6 | **8 ± 1.8** | 26.6 ± 6.6 | 0.71 ± 0.15 | 0.64 ± 0.18 | **0.87 ± 0.07** | 0.59 ± 0.20 | -1 ± 4.3 | -1.6 ± 6 | -2.7 ± 3.2 | 0.6 ± 10.1 |
| | 3–4 | 40 | 11.6 ± 2.9 | 19.6 ± 5 | 11.5 ± 2.8 | 31.8 ± 8.6 | 0.61 ± 0.20 | 0.43 ± 0.26 | 0.73 ± 0.15 | 0.42 ± 0.26 | -7.1 ± 4.6 | 8.4 ± 8.8 | 0.7 ± 4.9 | -14.3 ± 10.7 |
| SJ Encoder | 1–2 | 34 | 12.1 ± 3.3 | 11.6 ± 3.2 | 10.9 ± 2.9 | 22.0 ± 6.3 | 0.58 ± 0.23 | 0.54 ± 0.25 | 0.82 ± 0.12 | 0.28 ± 0.31 | 0.3 ± 5.6 | 0.4 ± 5.5 | 0.8 ± 5.1 | -0.1 ± 9.8 |
| | 3–4 | 23 | 8.7 ± 3.0 | 13.6 ± 4.7 | **5.9 ± 1.9** | 23.2 ± 8.4 | 0.63 ± 0.26 | 0.39 ± 0.36 | **0.92 ± 0.07** | 0.14 ± 0.42 | -1.3 ± 5.1 | 3.4 ± 8.1 | 2.0 ± 3.5 | -4.6 ± 12.2 |
| SJ Flight time | 1–2 | 47 | **5.6 ± 1.2** | 8.9 ± 2.0 | **4.8 ± 1.0** | 14.5 ± 3.3 | **0.89 ± 0.06** | 0.77 ± 0.13 | **0.96 ± 0.02** | 0.70 ± 0.15 | -0.8 ± 2.2 | -0.8 ± 3.5 | -1.6 ± 1.9 | -0.1 ± 5.6 |
| | 3–4 | 33 | **6.7 ± 1.8** | 11.5 ± 3.2 | **5.6 ± 1.5** | 18.6 ± 5.3 | **0.81 ± 0.12** | 0.68 ± 0.19 | **0.94 ± 0.04** | 0.58 ± 0.23 | -1.2 ± 3.2 | 3.7 ± 5.6 | 2.4 ± 2.8 | -4.7 ± 8.2 |
| Keiser leg press | 1–2 | 66 | **4.7 ± 0.9** | **5.1 ± 0.9** | **4.2 ± 0.8** | **9.0 ± 1.7** | **0.96 ± 0.02** | **0.83 ± 0.08** | **0.98 ± 0.01** | **0.91 ± 0.04** | 1.8 ± 1.6 | -0.4 ± 1.7 | 1.2 ± 1.5 | 2.2 ± 3.0 |
| | 3–4 | 45 | **4.1 ± 0.9** | **4.5 ± 1.0** | **4.0 ± 0.9** | **7.6 ± 1.7** | **0.97 ± 0.02** | **0.86 ± 0.08** | **0.98 ± 0.01** | **0.94 ± 0.04** | 0.3 ± 1.7 | 0.0 ± 1.9 | -0.2 ± 1.7 | 0.2 ± 3.1 |

Bold text denotes CV<10% and ICC>0.80. sample size in table. SJ: Squat jump, CMJ: Countermovement jump, $F_0$:Theoretical maximal force, $V_0$: Theoretical maximal velocity, $P_{max}$: Theoretical maximal power, $S_{FV}$: slope of the force-velocity profile.

**Table 5. Agreement and comparison for CMJ Force plate and SJ Force plate vs encoder, flight time and leg press measurements.**

| | | | Mean bias (±SD) | Mean bias % (±SD) | Standardized difference (±CL) | SEE (±CL) | SEE % (±CL) | Pearson r (±CL) | Slope of regression line | Y-intercept of regression line |
|---|---|---|---|---|---|---|---|---|---|---|
| **CMJ Force plate VS** | CMJ Encoder | $F_0$ (N) | 19 ± 233 | 1.2 ± 8.9 | 0.0 ± 0.2 | 238 ± 71 | 8.6 ± 2.7 | 0.865 ± 0.108* | 1.03 | -88 |
| | | $V_0$ (m/s) | -1.0 ± 0.5** | -22.8 ± 15.6 | -1.7 ± 0.3 | 0.5 ± 0.2 | 14.4 ± 4.6 | 0.508 ± 0.293* | 0.89 | 1.3 |
| | | $P_{max}$ (W) | -643 ± 248** | -22.2 ± 9.9 | -1.3 ± 0.2 | 243 ± 72 | 9.5 ± 3.0 | 0.878 ± 0.098* | 1.19 | 275 |
| | | $S_{FV}$ (N/m/s) | 256 ± 174** | 44.1 ± 25.5 | 1.3 ± 0.3 | 163 ± 49 | 23.2 ± 7.8 | 0.597 ± 0.258* | 0.64 | 110 |
| | CMJ Flight Time | $F_0$ (N) | 11 ± 180 | 0.0 ± 6.9 | 0.0 ± 0.2 | 152 ± 45 | 5.8 ± 1.8 | 0.947 ± 0.045* | 0.81 | 507 |
| | | $V_0$ (m/s) | -0.8 ± 0.5** | -19.3 ± 17.2 | -1.4 ± 0.3 | 0.5 ± 0.1 | 13.9 ± 4.5 | 0.562 ± 0.272* | 0.71 | 1.6 |
| | | $P_{max}$ (W) | 218 ± 199** | 31.4 ± 24 | 1.1 ± 0.4 | 126 ± 38 | 18.8 ± 6.2 | 0.783 ± 0.161* | 0.50 | 267 |
| | | $S_{FV}$ (N/m/s) | -550 ± 296** | -19.4 ± 13.3 | -1.1 ± 0.2 | 302 ± 90 | 12.2 ± 3.9 | 0.802 ± 0.149* | 1.00 | 545 |
| | Leg press | $F_0$ (N) | 415 ± 500** | 13.6 ± 17.8 | 0.9 ± 0.4 | 246 ± 73 | 9.5 ± 3.0 | 0.855 ± 0.115* | 0.48 | 1243 |
| | | $V_0$ (m/s) | -1.6 ± 0.6** | -34.8 ± 21.3 | -2.8 ± 0.4 | 0.6 ± 0.2 | 16.8 ± 5.5 | 0.106 ± 0.376 | 0.27 | 3.2 |
| | | $P_{max}$ (W) | -895 ± 253** | -30 ± 14.2 | -1.8 ± 0.2 | 255 ± 76 | 10.7 ± 3.4 | 0.865 ± 0.108* | 1.10 | 723 |
| | | $S_{FV}$ (N/m/s) | 764 ± 444** | 164.6 ± 42.7 | 3.8 ± 0.9 | 177 ± 53 | 26.4 ± 9.0 | 0.490 ± 0.299* | 0.19 | 460 |
| **SJ Force plate VS** | SJ Encoder | $F_0$ (N) | -194 ± 294** | -6.3 ± 10.9 | -0.4 ± 0.2 | 300 ± 89 | 10.3 ± 3.2 | 0.817 ± 0.140* | 0.96 | 310 |
| | | $V_0$ (m/s) | 0.0 ± 0.5 | 2.6 ± 21.7 | 0.1 ± 0.3 | 0.5 ± 0.1 | 19.9 ± 6.6 | 0.548 ± 0.278* | 0.93 | 0.2 |
| | | $P_{max}$ (W) | 215 ± 251** | 12.1 ± 12.4 | 0.5 ± 0.2 | 203 ± 60 | 11.1 ± 3.5 | 0.892 ± 0.088* | 0.72 | 350 |
| | | $S_{FV}$ (N/m/s) | -278 ± 327** | -19.4 ± 36.3 | -0.7 ± 0.3 | 331 ± 99 | 29.4 ± 10.2 | 0.569 ± 0.27* | 0.85 | 421 |
| | SJ Flight Time | $F_0$ (N) | -134 ± 400** | -4.4 ± 15.2 | -0.3 ± 0.3 | 389 ± 116 | 13.5 ± 4.3 | 0.662 ± 0.228* | 0.74 | 872 |
| | | $V_0$ (m/s) | 0.2 ± 0.6** | 11.4 ± 28 | 0.4 ± 0.4 | 0.5 ± 0.2 | 22.8 ± 7.7 | 0.405 ± 0.325* | 0.47 | 1.2 |
| | | $P_{max}$ (W) | 99 ± 236** | 5.8 ± 13.2 | 0.2 ± 0.2 | 224 ± 67 | 12.4 ± 4.0 | 0.866 ± 0.106* | 0.82 | 244 |
| | | $S_{FV}$ (N/m/s) | -186 ± 422** | -12.5 ± 51.2 | -0.5 ± 0.4 | 394 ± 117 | 36.1 ± 12.9 | 0.207 ± 0.366 | 0.32 | 899 |
| | Leg press | $F_0$ (N) | 238 ± 704 | 6.0 ± 28.9 | 0.5 ± 0.5 | 437 ± 130 | 15.4 ± 5.0 | 0.541 ± 0.281* | 0.33 | 1877 |
| | | $V_0$ (m/s) | -0.3 ± 0.7** | -11.7 ± 34.7 | -0.6 ± 0.4 | 0.6 ± 0.2 | 24.0 ± 8.1 | -0.177 ± 0.370 | -0.45 | 3.5 |
| | | $P_{max}$ (W) | -136 ± 187** | -7.2 ± 10.6 | -0.3 ± 0.2 | 191 ± 57 | 10.1 ± 3.2 | 0.905 ± 0.078* | 1.03 | 95 |
| | | $S_{FV}$ (N/m/s) | 276 ± 665** | 23.5 ± 84.5 | 0.7 ± 0.7 | 401 ± 120 | 37.2 ± 13.3 | -0.074 ± 0.378 | -0.06 | 1327 |
| | CMJ Force plate | $F_0$ (N) | -177 ± 424** | -5.9 ± 16.5 | -0.3 ± 0.3 | 406 ± 121 | 14.0 ± 4.5 | 0.623 ± 0.246* | 0.68 | 1042 |
| | | $V_0$ (m/s) | 1.3 ± 0.8** | 70.0 ± 34.7 | 2.2 ± 0.6 | 0.6 ± 0.2 | 24.6 ± 8.3 | -0.015 ± 0.380 | -0.02 | 2.5 |
| | | $P_{max}$ (W) | 759 ± 306** | 54.8 ± 15.7 | 1.7 ± 0.3 | 274 ± 82 | 14.9 ± 4.8 | 0.793 ± 0.155* | 0.70 | 1.0 |
| | | $S_{FV}$ (N/m/s) | -488 ± 423** | -32 ± 62.9 | -1.2 ± 0.4 | 400 ± 119 | 37.1 ± 13.3 | 0.105 ± 0.376 | 0.21 | 1083 |

Sample size = 27

*Significant correlations p<0.05

**Significantly different from comparison measure (SJ/CMJ force plate) p<0.05. SJ: Squat jump, CMJ: Countermovement jump, SEE: Standard error of estimate. SD: Standard deviation, CL: 95% Confidence limit.

for $V_0$ (and thereby $S_{FV}$) during vertical jumping could be a consequence of calculating velocity from a force signal (force plate). However, our data show low reliability for $V_0$ from CMJ and SJ regardless of the velocity calculation method. The velocity from the leg press exercise is calculated as the derivation of position over time, identical to the encoder during SJ and CMJ, making it less likely that the variation in $V_0$ is caused by calculation error. Further, Meylan et al. [12] speculated that the low $V_0$ reliability is caused by greater biological variation closer to $V_0$. However, our data show similar typical errors across loads and similar typical errors for $F_0$ and $V_0$ from the leg press (using loads with similar distance to both intercepts), making this questionable.

Furthermore, García-Ramos et al. [26] showed that the low $V_0$ reliability during vertical jumping was most likely due to the distance of the extrapolation to the $V_0$ intercept [26], as the lightest load possible to assess is the subject's own bodyweight. The influence of the

**Table 6. FV-variables for all methods.**

| | $F_0$ (N) | $V_0$ (m/s) | $P_{max}$ (W) | $S_{FV}$ (N/m/s) |
|---|---|---|---|---|
| **CMJ Force plate** | 2741 ± 491 | 3.8 ± 0.7 | 2537 ± 527 | 771 ± 260 |
| **CMJ Encoder** | 2760 ± 415 | 2.8 ± 0.4 | 1906 ± 360 | 1016 ± 225 |
| **CMJ Flight time** | 2759 ± 549 | 3.1 ± 0.6 | 2090 ± 380 | 948 ± 346 |
| **SJ Force plate** | 2915 ± 561 | 2.5 ± 0.7 | 1806 ± 464 | 1249 ± 483 |
| **SJ Encoder** | 2621 ± 404 | 2.5 ± 0.4 | 1652 ± 361 | 1065 ± 244 |
| **SJ Flight time** | 2794 ± 476 | 2.7 ± 0.5 | 1925 ± 498 | 1059 ± 270 |
| **Keiser leg press** | 3156 ± 831 | 2.1 ± 0.2 | 1660 ± 389 | 1519 ± 510 |

Sample size = 27. SJ: Squat jump, CMJ: Countermovement jump, $F_0$:Theoretical maximal force in newtons, $V_0$: Theoretical maximal velocity in meters per second, $P_{max}$: Theoretical maximal power in watts, $S_{FV}$: slope of the force-velocity profile. Values are presented as mean ± standard deviation.

extrapolation distance has been discussed earlier [25], and the present results reinforce this assumption. $F_0$ and $V_0$ displayed similar reliability in the leg press exercise as the loads approached both ends of the FV-spectrum. The high reliability in the FV-variables obtained from the leg press can also partly be attributed to better standardisation in terms of fixed seat position, and thereby less technical variation in the exercise execution compared to the free weight conditions during CMJ and SJ [17, 18, 44, 45]. The influence of standardisation is also supported by the findings of Valenzuela et al. [19], which showed superior reliability of the FV variables obtained using a smith machine compared to free weights. It is therefore likely that the observed variations in the extrapolated variables $V_0$ and $S_{FV}$ are caused by extrapolation error (i.e., small variations in the individual attempts are amplified because of the "extrapolation distance") and the combination of technical/instrumental and biological variations. Consequently, in addition to superior standardisation compared to the other tests, the larger load range in the leg press exercise reduces the need for extrapolation for both force and velocity, explaining the high reliability of all the FV variables (Table 7).

The FV variables showed some slight differences in reliability between the CMJ and SJ conditions (Table 3). These small differences can partly be explained by slope steepness differences between SJ and CMJ, as the extrapolation distance to each intercept varies between these conditions (Table 7 and Fig 4). Additionally, SJ is prone to integration errors when calculating velocity with the force plate method [29]. This is linked to the assumption of zero start velocity, which is technically more challenging during SJ compared to CMJ. This challenge is similar for the encoder method, as the average force and velocity are calculated at the instance of the encoder's registration of a positive displacement. These issues are reinforced by the fact that the flight time method showed the highest reliability for all FV-variables in SJ compared to the other methods (Table 3). Hence, the poor reliability of the SJ force plate and encoder method may be explained by calculation errors rather than physiological differences between the CMJ and SJ condition. Consequently, when calculating FV-profiles from encoders and force plates during SJ, careful attention should be given to the pause at the bottom (static position) of the squat to improve the detection of movement with this equipment (i.e., giving athletes extra practice attempts and/or familiarization).

Interestingly, the FV-variables measured with the encoder during CMJ exhibited the lowest CV% of all the CMJ measurement methods during the vertical jumps (Table 3). Notably, the encoder software uses the entire positive displacement curve, including the airtime. Additionally, average force is calculated as the product of mass and acceleration, where acceleration is the average velocity divided by the duration of the positive displacement. Especially in light loading conditions where the flight time is relatively long, changes and variability in force or

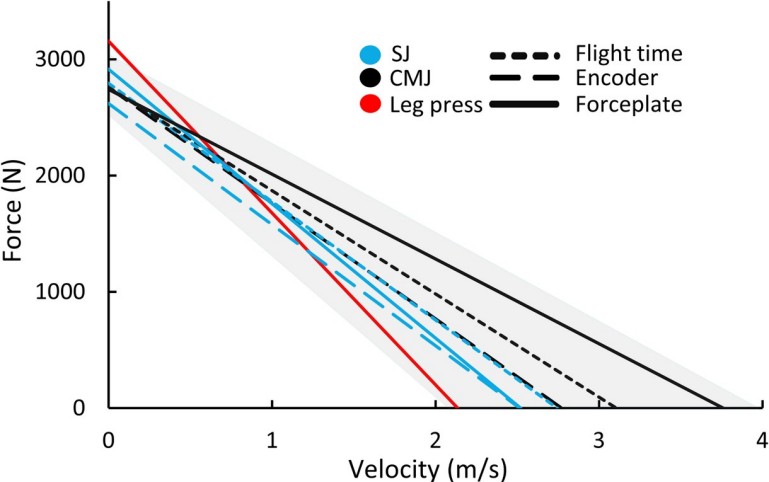

**Fig 4. Shows averaged force-velocity profiles from all methods for the main analysis (n = 27).** The shaded area represents the 95% confidence interval for the vertical jumps.

velocity for the propulsive phase are inevitably harder to detect. Although the software manufacturer uses these calculations to improve reliability, the validity of the FV-profile will also be affected, considering the ability to detect changes. Additionally, changes in the estimated force in the light loading conditions are proportionally more affected by changes in bodyweight than changes in propulsive force (when the flight phase is greater than the push-off phase). With lower flight times, the encoder's measures will to a greater degree reflect changes in propulsive force. This is supported by the correlation of 0.86 for $F_0$ between the force plate method and the encoder. The greater reliability observed for the FV-variables assessed by the encoder may be misleading, as the usefulness of a test is determined not only by reliability and validity, but also by the ability to detect changes in performance [10].

The reliability results for the force plate method and flight time method were practically identical for all FV-variables during CMJ, but not SJ (Table 3). The differences between the force plate method and flight time method for SJ were probably due to the difficulty of detecting the zero starting velocity in the SJs for the force plate method, as discussed earlier [29]. This contention is supported by the fact that both methods (flight time and force plate method) showed similar reliability in the CMJ, as the zero starting velocity issue is not present in the CMJ. Furthermore, the slightly better reliability in SJ for the flight time method

**Table 7. Loading ranges used to assess the force velocity profiles.**

| | Force in % of $F_0$ | | Velocity in % of $V_0$ | |
|---|---|---|---|---|
| | *Heaviest load* | *Lightest load* | *Heaviest load* | *Lightest load* |
| *CMJ Force plate* | 75 ± 6 | 56 ± 6 | 26 ± 6 | 46 ± 7 |
| *CMJ Encoder* | 63 ± 6 | 39 ± 6 | 37 ± 6 | 61 ± 6 |
| *CMJ Flight time* | 75 ± 7 | 56 ± 6 | 25 ± 7 | 46 ± 9 |
| *SJ Force plate* | 68 ± 10 | 50 ± 8 | 33 ± 9 | 56 ± 14 |
| *SJ Encoder* | 66 ± 7 | 37 ± 6 | 35 ± 7 | 63 ± 5 |
| *SJ Flight time* | 70 ± 10 | 52 ± 8 | 32 ± 9 | 58 ± 15 |
| *Keiser leg press* | 80 ± 9 | 18 ± 3 | 22 ± 8 | 84 ± 4 |

Sample size = 27. SJ: Squat jump, CMJ: Countermovement jump, $F_0$:Theoretical maximal force, $V_0$: Theoretical maximal velocity. Values are presented as mean ± standard deviation.

compared to the CMJ condition was probably due to less variation in starting position, as this is easier to control with the pause at the bottom of the squat.

Conjointly, the reliability of $F_0$, $V_0$ and $P_{max}$ was affected by the variation in the measurements–of each individual load–combined with the degree of extrapolation to the FV-intercepts. Hence, $S_{FV}$ was inevitably affected by the variation in both $F_0$ and $V_0$. Researchers and coaches should be aware of these limitations when assessing individual FV-profiles. Indeed, the 5–10% CV in jump height observed in this study was not acceptable for accurately assessing the accompanying FV-variables $V_0$ and $S_{FV}$, regardless of measurement method, with a loading range of bodyweight to 80 kg (forces ranging from 40–70% of $F_0$). Typical error can only be decreased by reducing the variation in jumping performance or including loads closer to the $F_0$ and $V_0$ intercept. Additionally, the usefulness of a test is determined by the ability to detect changes in performance; more specifically, by comparing the typical error (CV%) with SWC [46]. Indeed, the FV-variables obtained from the leg press apparatus showed a superior signal-to-noise ratio compared to the other measurement methods in this study (Fig 3).

## Agreement among methods

Calculating the velocity of the center of mass from ground reaction forces has previously shown comparable reliability, with only small measurement errors compared to the "gold standard" 3D motion capture systems [47, 48]. It can therefore be argued that the force-plate method is the most valid method for assessing FV-profile during vertical jumping compared to all other measurement methods used in this study.

Only a few studies have examined the relationships among varying FV-profile methods for the lower limbs. García-Ramos et al. [22] also observed strong correlations for $F_0$ and $P_{max}$ and trivial correlations for $V_0$ and $S_{FV}$ across methods (force plate, linear encoder and flight time methods). Similar to the present study, the poor agreement for $V_0$ and $S_{FV}$ was explained by the large extrapolation error for $V_0$ [22].

Contrary to our findings, Jiménez-Reyes et al. [15] reported excellent agreement between the flight time and force plate method for the FV-variables (r: 0.98–0.99). This discrepancy from our findings can probably be attributed to several methodological differences. The flight time method calculates force and velocity based on jump height [15]. However, flight times are inevitably prone to small errors in technical execution [49], in addition to systematic errors compared to jump height obtained from force data [50, 51]. As Jiménez-Reyes et al. [15] point out, the FV-variables are associated with cumulative extrapolation errors, consecutively decreasing the validity of these variables. The small systematic and random differences in jump height between flight time and force data are even greater for the extrapolated FV-variables. Additionally, the assumption of constant acceleration during the push-off phase in the flight time method could also affect the agreement with the force plate method, as variations in average force and velocity during the push-off phase are not necessarily related to jump height variations [17, 18, 52, 53].

Furthermore, the flight time method assumes constant push-off distance across loads and trials [15, 31]. However, from the force plate data, we observed 5–10% (2–4 cm) variation in push-off distance across trials and loading conditions, even when controlling the depth as previously recommended [54]. This variation may be due to changes in jump mechanics across trials and loads [45], making it challenging to assume a constant push-off distance despite controlled knee angle. Jiménez-Reyes et al. [15] have previously reported a 0.4% variation (CV%) in push-off distance across trials for CMJ when using a smith machine. This apparatus probably reduces the variation in jump mechanics compared to the free weight jumps used in the present study. This implies that the poor agreement in our study can also be attributed to poor control of the center of mass for the subject, and not solely the flight time method.

Contrary to previous research showing an overestimation of $V_0$ measured with an encoder compared to a force plate (72.3%) [22, 47], we observed an underestimation for the CMJ condition (-23%) (Table 6). The overestimations of velocity during light loading conditions in previous investigations are explained by the attachment point at the bar, as the bar velocity is higher than the centre-of-mass velocity during jumping [22, 47]. However, because the velocity from the encoder used in this study is based on the entire positive displacement curve (including the airtime), the average velocity is lower. Combined with the extrapolation error, this partly explains the higher agreement between the force plate and encoder for $F_0$ and $P_{max}$ compared with $V_0$ and $S_{FV}$. Practitioners and researchers should be aware of the limitations of using linear encoders for measuring FV-profiles, especially to obtain $V_0$ and $S_{FV}$.

Padulo et al. [21] observed an underestimation in $V_0$ (-46%) and overestimation in $F_0$ (21%) in the leg press compared to the squat exercise. The underestimation in $V_0$ can be attributed to biomechanical differences, as the squat movement involves a larger contribution from the hip joint, resulting in higher system velocity [21]. In addition, approximately 30% of the work during a vertical jump is contributed by the ankle joint [45]. This contribution is likely lower for the leg press due to the more plantarflexed orientation of the ankles in this apparatus. These biomechanical differences probably explain why the leg press has the largest bias of all the tested methods (Table 6). Another explanation is the pneumatic resistance in the present leg press apparatus, allowing higher average velocities for a given force due to the absence of inertia [55]. Additionally, the software excludes 5% of the range of motion from the start and end of the movement, inevitably affecting the average values in the lighter resistance conditions to a greater degree compared to the higher resistance conditions, resulting in higher $V_0$. These issues may explain the high $V_0$ in the leg press exercise and the low agreement in $V_0$ compared to the other measurement methods. Intriguingly, $V_0$ was negatively correlated with the three SJ measures and the leg press exercise (Fig 5). The extrapolated $V_0$ during the leg press exercise is highly influenced by the push-off distance [56], where it has been previously argued that comparisons across individuals should only be done when participants perform the vertical jumps with their usual or optimal push-off distance Samozino et al. [57]. The initial push-off distance during vertical jumping in this study was self-determined, while the push-off distance in the leg press was standardised, possibly explaining the poor correlation in $V_0$ between the leg press and the jump exercises. Furthermore, as shown by Bobbert [56], the linear shape of the FV-relationship during multi-joint movements is influenced by segmental dynamics, and this influence is magnified by increasing movement velocity [56]. Hence, segmental dynamics probably influence the agreement of $V_0$ to a greater degree than $F_0$ when comparing exercises with varying push-off distances and joint contributions [56]. Consequently, segmental dynamics partly explain the larger agreement for measures closer to $F_0$ and poorer agreement and correlations for $V_0$ across leg press and vertical jump tasks. As illustrated in Fig 4 and shown in Table 5, differences in $V_0$ are larger across methods and conditions compared to $F_0$.

Small but important differences across methods accumulate, with larger differences for $V_0$ and $S_{FV}$ compared to $F_0$ and $P_{max}$. The agreement across methods is highly influenced by the combination of measurement errors, as well as the distance of extrapolation to the FV-intercepts. All FV-variables depend on the measurement condition, including equipment, exercise type, resistance modality and push-off distance.

## Strengths and limitations

The present study included a large sample of male and female athletes with varying sport backgrounds, using a multicenter approach. This design allows for larger sample sizes and higher ecological validity as athletes are assessed by different test leaders and using different

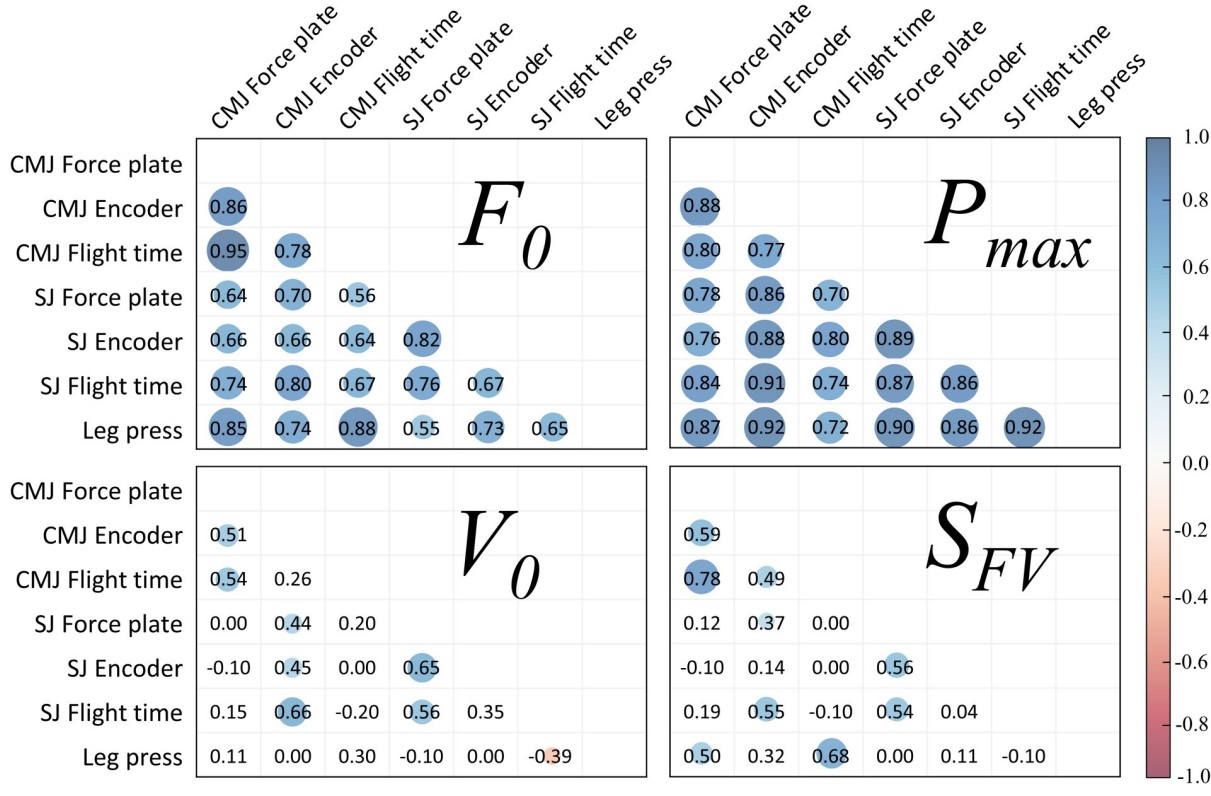

**Fig 5. Correlation matrix showing Pearson r coefficients for the FV-profile variables ($F_0$, $P_{max}$, $V_0$, $S_{FV}$) for cross sectional data.** Colored circles indicate $P<0.05$, where circle size and color represent corresponding r values (color legend is presented with the figure). SJ: Squat jump, CMJ: Countermovement jump, $F_0$: Theoretical maximal force, $V_0$: Theoretical maximal velocity, $P_{max}$: Theoretical maximal power, $S_{FV}$: slope of the force-velocity profile. Sample size for all correlations n = 27.

equipment [58]. The conclusions from the study are based on the results from the main analysis and supported by the data from the larger aggregated analysis.

There are several methodological limitations that need to be considered for the findings from this study. The difference in number of loading conditions (i.e., 5 for vertical jumping and 10 for leg press) and relative position on the FV-curve inevitably affect the agreement measures due to differences in the accuracy of obtaining the extrapolated variables. Additionally, the difference in push-off distance from the leg press (standardized to vertical femur) and vertical jumping (standardized to self-selected depth) may influence the variation across these conditions. The leg press protocol included breaks of 10–20 sec for the light loads and 20–40 for the heavy loads, which may cause some fatigue between repetitions and influence the FV-relationship. For the force plate method, the 5 SD threshold for determining the start of the movement will influence the average values of force and velocity and thereby the FV-variables. Especially in the SJ, but also in the CMJ, this threshold is sensitive to small movements and is a source of error that is not controlled for. In the leg press software, the average values have a 5% cut-off from the range of movement, which can lead to i) taller athletes having a larger cut-off in terms of absolute values compared to shorter athletes, and ii) in the lighter loads where more range of motion is achieved, the cut-off in terms of absolute values will be larger for lighter loads compared to heavier loads. The results from the encoder used in the present study cannot be generalized to other linear encoder devices with different calculation methods for acceleration and force. The jumps in this study were performed with free weights, where it was

difficult to accurately standardize the center of mass of the jumps using only thigh depth or knee angle as a reference. These variations in the center of mass are likely smaller using smith machines. These limitations inevitably affect both the test-retest reliability and the agreement across methods, where it is impossible to differentiate which source of variability leads to the results observed in this study. Nevertheless, the use of free weights increases the ecological validity of the study as these are commonly used by athletes. Additionally, for the analysis for agreement the force plate was sampled at 200 Hz compared to 1000 Hz used previously [15], which may have influenced the findings. For the aggregated reliability analysis, both 200 Hz and 2000 Hz force plates were used, and we would argue that the findings of reliability seem independent of sampling frequency.

## Conclusions and practical applications

A 5–10% between-session CV in jump height is not acceptable for accurately assessing $S_{FV}$ and $V_0$, regardless of measurement method, using a loading range of bodyweight up to 80 kg (forces ranging from 40–70% of $F_0$). Caution is advised when using similar protocols for individual training recommendations or interpreting training adaptions for athletes. Efforts should be made to either reduce the variation in jumping performance or to assess loads closer to the FV-intercept. Increasing the loading range can be achieved by using alternative exercises such as a leg press exercise. Reducing the variation in jumping performance may possibly be achieved through additional practice attempts, and attention should be given to the depth of the squatting motion during the vertical jumps. $F_0$ and $P_{max}$ showed high reliability and generally good agreement across measurement methods, indicating that these variables can be used with confidence by researchers and coaches. However, one should be aware of the poor reliability of the FV-variables $V_0$ and $S_{FV}$ obtained from vertical jumping, as well as differences across measurement methods for assessing individual FV-relationships.

## Supporting information

**S1 Dataset.**
(XLSX)

## Acknowledgments

We would like to thank all athletes who participated in the present study.

## Author Contributions

**Conceptualization:** Bent Rønnestad, Gøran Paulsen.

**Data curation:** Thomas Bjørnsen, Christian Helland, Thomas Haugen.

**Formal analysis:** Thomas Bjørnsen, Christian Helland, Bent Rønnestad, Gøran Paulsen.

**Funding acquisition:** Bent Rønnestad, Gøran Paulsen.

**Investigation:** Kolbjørn Lindberg, Christian Helland, Bent Rønnestad, Martin Thorsen Frank, Thomas Haugen, Sindre Østerås, Morten Kristoffersen, Magnus Midttun, Fredrik Sæland, Gøran Paulsen.

**Methodology:** Paul Solberg, Christian Helland, Bent Rønnestad, Martin Thorsen Frank.

**Project administration:** Thomas Bjørnsen, Bent Rønnestad, Thomas Haugen, Gøran Paulsen.

**Resources:** Thomas Bjørnsen, Bent Rønnestad, Gøran Paulsen.

**Supervision:** Paul Solberg, Thomas Bjørnsen, Bent Rønnestad, Gøran Paulsen.

**Validation:** Gøran Paulsen.

**Visualization:** Kolbjørn Lindberg.

**Writing – original draft:** Kolbjørn Lindberg, Martin Thorsen Frank.

**Writing – review & editing:** Kolbjørn Lindberg, Paul Solberg, Thomas Bjørnsen, Christian Helland, Bent Rønnestad, Martin Thorsen Frank, Thomas Haugen, Sindre Østerås, Morten Kristoffersen, Magnus Midttun, Fredrik Sæland, Gøran Paulsen.

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
