## [Decision Letter · Decision Letter 0]

9 Nov 2020

PONE-D-20-29309

Force-velocity profiling in athletes:

Reliability and agreement across methods.

PLOS ONE

Dear Dr. Lindberg,

Thank you for submitting your manuscript to PLOS ONE. After careful consideration, we feel that it has merit but does not fully meet PLOS ONE’s publication criteria as it currently stands. Therefore, we invite you to submit a revised version of the manuscript that addresses the points raised during the review process.

We look forward to receiving your revised manuscript.

Kind regards,

Daniel Boullosa

Academic Editor

PLOS ONE

Journal Requirements:

2. Please include captions for your Supporting Information files at the end of your manuscript, and update any in-text citations to match accordingly. Please see our Supporting Information guidelines for more information: http://journals.plos.org/plosone/s/supporting-information

Reviewers' comments:

Reviewer's Responses to Questions

**Comments to the Author**

1. Is the manuscript technically sound, and do the data support the conclusions?

Reviewer #1: Yes

Reviewer #2: No

2. Has the statistical analysis been performed appropriately and rigorously? 

Reviewer #1: Yes

Reviewer #2: Yes

3. Have the authors made all data underlying the findings in their manuscript fully available?

Reviewer #1: Yes

Reviewer #2: Yes

4. Is the manuscript presented in an intelligible fashion and written in standard English?

Reviewer #1: No

Reviewer #2: Yes

5. Review Comments to the Author

Reviewer #1: The present study assessed the between-day reliability of the f-v parameters computed from different exercises (CMJ, SJ and leg press) and using different methods (force plate, encoder, flight time), as well as the differences in the magnitude of the f-v parameters assessed with these methods. Additionally, the authors examined the association of these parameters with some performance measures.

I think the topic is relevant, particularly given the growing popularity of these methods for guiding training prescription and for performance assessment. Moreover, the results are quite novel and could have practical implications. My main concern is that the manuscript (including the text, figures and tables) is a bit difficult to follow in some parts, and it could maybe be simplified to enhance readability. Moreover, some typos should be corrected.

Specific comments:

Abstract

- Line 24. The athletes ‘were’ tested…?

- Line 41. I suggest including in the last sentence of the abstract that coaches should also be aware of the low reliability of f-v parameters (particularly v0 and Sfv) assessed during jumping.

- There is no comment on the association between fv parameters and performance measures. In my opinion, and just as a suggestion that the authors are free to obviate, the manuscript is strong enough when presenting reliability and magnitude comparison methods, and adding the association between fv parameters and performance measures complicates the manuscript. This last section, which is presented as a secondary aim of the study, could be presented even in a separate paper, but this is just in my humble opinion.

Introduction

- It would be nice to cite this review, which gives a nice overview of the applications of VBRT (and of the assessment of the fv profile): https://journals.lww.com/nsca-scj/Abstract/9000/Velocity_Based_Training__From_Theory_to.99257.aspx

- Line 50. It would be nice to mention that there is some controversy regarding the linearity of the f-v relationship, at least at very low force values (PubMed ID: 32255757, PubMed ID: 26103786).

- Line 57. Can the authors include a reference for this equation? Maybe one of the studies/reviews of Morin and Samozino would be nice.

- Line 60. Please, specify how these f-v parameters were computed. From jump height? From force platforms? linear encoders? Has the between-day reliability of the f-v profile estimated from jump height been proven?

- Line 65. Please, specify which measurement methods.

- Line 69. Please, specify the method of measurement. Jump height? Force platform?

- Line 72. A recent study by Valenzuela et al (accepted in IJSPP, but still in press) reported a low between-day reliability for f-v parameters computed from jump height during vertical jumps. Please, see attached document. It would be nice to briefly mention this article in the introduction or discussion section.

- Line 74. Should it be "there exist numerous methods..."?

- Line 87. "...as well as from the leg press exercise".

- Line 88. Previous research "has" investigated...

Methods

- Line 134. Which was their experience with the testing procedures?

- Line 167. It would be nice to include a new paragraph with each test, as done for SJ and CMJ.

- How much time was left between tests?

- Line 202. It would be nice to start a new paragraph with each measurement method (as done for the Keiser leg press)

Results

- Line 259. coma not needed before ‘ranging’.

- Line 269. 1.2 cm for both SJ and CMJ?

- There is no mention to the SWC in the results section. Were differences between days or between methods greater than the SWC?

- Line 291. The results on the association between fv parameters and performance measures should be explained in greater detail in the text (if the authors want to keep these analyses). On which tests (CMJ, SJ, leg press) was an association found? With which measurement method? (encoder, force plate, flight time).

Discussion

In my opinion the discussion section could be shortened and simplified. The authors could try to avoid repeating concepts.

- Line 298. ‘were’ unreliable? The authors are mentioning two variables.

- Line 332 and Line 423. Small differences in starting position during jumping (impulse distance) can have meaningful effects on the fv variables (PMID: 32223526).

- Line 397. ‘has’ previously shown…

- Line 509. Values for analysis "are" used

- Line 510. the observation ... has, or the observations .... have..

- Line 511. that ‘appears’ when using…

- Line 517. less error for V0? Wouldn’t this error be larger given that the relationship losses linearity particularly at values closer to v0?

- Line 534. push-off instead of push of.

- Line 547. The jumps…’were’

- Line 549. These variations…’are’.

Conclusions

- Line 55. Specify that this CV is between-session

Tables 3-5. Just in case it is possible. Could the tables be simplified assessing between-session reliability pooling all subjects together? Without dividing tests 1-2 and 3-4.

Table 6. Please, specify what methods are being compared. What does the bias refer to in each line? CMJ vs leg press? CMJ vs SJ? Encoder vs flight time? Where is the comparison of leg press vs SJ flight time, for instance?

Table 6. Footnote. Units are already specified in the table, no need to explain them in the footnote.

Table 6. Footnote. ‘significant differences’, or ‘significantly different’? Significant different would be incorrect.

Table 7. Please specify the units for F0, v0, Pmax and Sfv

Figure 2. tested instead of tester. Also, please, include the "n" always in capital or lowercase.

Figure 3. Panel B. The Y axis can be reduced (the maximum value is lower that 10-15). Also, the SWC is not discussed in the results section, and almost not discussed in the discussion section. I suggest either discussing this result in greater detail, or removing it. Were differences between days or between methods greater than the SWC?

Figure 5. If the aim is to assess the association between f-v parameters and performance measures, could this figure be simplified by removing the analyses on the left? The authors would just need to show the association between F0, v0, Pmax and Sfv with 1RM, CMJ, SJ, 10m sprint and 30m sprint, but there is no need to show the association between f0 leg press and f0 CMJ force plate, for instance.

Reviewer #2: Manuscript ID: PONE-D-20-29309

GENERAL COMMENTS: First, I would like to congratulate the authors on their effort. The study comprises a large sample given the elite level of the participants. The fact that several training centers participated on the research adds value to it. The aim of the study was to investigate the test-retest reliability and agreement across methods for assessing individual Force Velocity-profiles of the lower limbs in well-trained athletes. The research idea is interesting, and the conclusions may have important implications in current athlete profiling procedures. The investigation has a sound scientific background given the previously questioned reliability of the FVP parameters. However, some methodological aspects must be polished before I can recommend the publication of the study. I believe that some changes are necessary to strengthen the quality of the data reported and the conclusions. I do think that the data collected may positively impact the sport science field. The authors should interpret my comments as constructive.

Abstract

Changes in the abstract will not be addressed prior to all the other issues within the manuscript are solved.

Introduction

In general, the introduction tries to explain and introduce the research problem and is fairly successful. It is clear for the reader the need to conduct the present investigation.

Specific comments:

Line 59-60. "However, although several studies have evaluated the between-session reliability of FV-parameters". In general terms, what did these studies found? It would be interesting for the reader if the authors presented a bit more detail here.

Methods

The methods are interesting but that are some details that must be addressed.

Specific comments:

Line 103-105. “The first two (...) figure 1 & 2)". This sentence is somehow confusing. Based on the information presented in Figure 1, the authors should consider rephrasing as follows:

"The first two testing timepoints were separated by ~1 week, before a period of 2~6 months. Then, the two-last testing timepoints were conducted also were separated by ~1 week (figure 1 & 2)."

Line 114. Please consider replacing "are" before "constant" with "were".

Line 116. Please consider replacing "was" before "constant" with "were kept".

Line 133. "national and elite level". What did the authors consider to be national and elite level? A better description of the criteria used to classify the athletes would be interesting for the reader.

Line 143. "hamstring and hip mobility". What do the authors mean by hamstring mobility? Please clarify.

Line 150-151. "for some weaker subjects, a protocol of approximately 5 loads up to 80% of bodyweight were used". Was there a specific criteria to determine this? Which athletes were considered as "weaker subjects"? And the 5 loads, were they standardized or individually determined? Please clarify.

Line 161. Please consider replacing "was verbally forbidden" by "was not allowed".

Line 167. Please consider starting a new paragraph with "For the leg press,...". Otherwise, the paragraph will be too long and hard to follow.

Line 168-170. “(...) the FV-parameters were derived from a dedicated software based on a 10-repetition FV-test with incremental loading based on each athlete's 1RM load". This sentence is confusing. Please consider rephrasing.

Line 171. “∼20% of 1RM”. The 1RM load was previously determined? Or was the athlete's perceived 1RM used?

Line 177. It is spelled "heels" and not "heals". Please correct.

Line 179. Please consider replacing "are" before "performed" with "is".

Line 183. Please consider starting a new paragraph with "Prior to the 30-m sprint, …".

Line 183. Did all participants perform the 30 m sprint test or just the athletes from running-based sports (e.g., soccer, handball or athletics)? I mean, did the weightlifters, ski-jumpers or badminton players also performed a 30-m sprint?

From what I understand one of the aims of the study is to "investigate the association between the FV-parameters obtained from vertical jumping and leg press with 1RM squat, jump height and 10 and 30m sprint time". I see no problem with investigating the association between the FV parameters and 1RM squat and jump height for the entire sample. However, when it comes to the sprint test, I am not sure that the analysis should include the weightlifters or sky-jumpers, for example. Do the authors consider that any meaningful practical application can be drawn from a ski-jumper's 30-m sprint performance?? If these athletes were included in the analysis for the sprint variables, I strongly recommend the authors to exclude them from this outcome. Also, I strongly suggest doing a sub-group with only the athletes from running-based sports (i.e., soccer, handball, ice-hockey, speed skating and possibly athletics - depending on the modality) and test the associations between FV parameters and 1RM, vertical jump and sprint outcomes. For the rest of the sample, I suggest testing the relationship between the FV parameters and only the 1RM and vertical jump (exclude sprint).

This will greatly strengthen the quality and, especially, the logic behind the study as it will provide meaningful data for practitioners, based on the specific physical performance variables from each sport without inducing them to perform unnecessary and “injury risk tests” (e.g., maximal sprint test for a weightlifter or a sky-jumper).

Line 188. Please consider starting a new paragraph with "The 1RM back-squat…".

Line 230. The fact that one force plate recorded at 200 Hz while other at 2000 Hz must be acknowledged as a limitation that may potentially affect data analysis and comparison between methods.

Results

The figures are well designed and facilitate the interpretation of the data.

Line 291. As stated before, I strongly recommend the authors to exclude the athletes from non-running based sports from the sprint analysis to strengthen the conclusion and ecological validity of the data reported in the study.

Discussion

Overall, the discussion is well written, and the authors do a good job comparing the results obtained with previous research. They present a sound reasoning for their findings, supporting their data adequately based on previously published literature. However, the section is too lengthy and would greatly benefit if the authors were able to be more concise with their writing and reduce the word count. Some of the paragraphs repeat information, which makes this section hard to follow at times. I strongly recommend the authors to invest time re-arranging the discussion because I do believe that the paper is interesting for the sport science community and has the potential to be published and impact current athlete profiling practices.

Specific comments:

Line 306-334. This paragraph is too long. I recommend the authors to shorten it or divide it into multiple paragraphs.

Line 334. All tables should be presented in the Results section. I recommend the authors to move Table 8 into the mentioned section.

Line 353. According to the table presented in Table 7, the Pmax obtained in the SJ was higher than the CMJ when using the encoder. However, in the other methods, the Pmax was higher in the CMJ (as it would be expected). The authors must explain these contradicting results in this paragraph.

Line 378-385. The information presented in this paragraph have been previously presented in the discussion section (Line 319, 322-327). The authors should consider removing this part of the text to avoid repetition and improve the "fluidity" of the discussion.

Line 397. Please consider replacing "have" before "previously" with "has".

Line 407-437. This paragraph is too long. I recommend the authors to shorten it or divide it into multiple paragraphs.

Line 479. "All FV-variables depend on the measurement condition, including equipment, exercise type, resistance modality and push-off distance." What are the practical applications from this? I believe the discussion could be shortened and provide a more "applied" perspective.

Line 483-521. This paragraph is too long. I recommend the authors to shorten it or divide it into multiple paragraphs.

Line 509. Please consider replacing "is" before "used" with "are".

Line 532. Strengths and limitations. Another important limitation is related to the fact that, for the aggregated analysis, force plates with different sampling frequencies were used. This must be acknowledged in the manuscript and its implications for the results must be briefly addressed (one sentence would suffice).

Line 526. The argument used regarding the "ecological validity" of the study is greatly affected by the fact that athletes whose training regimens do not usually incorporate linear sprint actions (e.g., weightlifting or badminton), where tested for 30-m sprint performance. For this reason, the authors should re-consider the analysis made. The data presented herein is really interesting and address important aspects related to athlete profiling. I consider that publishing such data would be important for the sport science community. It would be relevant, for example, to include some discussion related to this issue, emphasizing that athletes from non-running/sprint-based sports should not perform sprint-based tests, as this is “not rational from a logical standpoint”. Therefore, I strongly advise making the previously mentioned adjustments as it would greatly improve the quality of the data.

Line 574. Please consider replacing "was" before "performed" with "were".

547-549."The jumps in this study were performed with free weights where it was difficult to accurately standardize the center of mass of the jumps using only thigh depth or knee angle as a reference." Although I understand what the authors mean here, I am not sure if performing the exercises with free weights is an actual limitation of the study. Most athletes train with free-weights, so that increases the "ecological validity" of the data reported. Moreover, it is really important for coaches to understand that calculating FV-parameters using free weights affects test-retest reliability due to the reasons mentioned throughout the manuscript. It is not a limitation, but rather an “unavoidable” phenomenon that occurs is real-world training settings.

Conclusions and practical applications

The conclusions seem appropriate and based on the results obtained.

6. PLOS authors have the option to publish the peer review history of their article (what does this mean?). If published, this will include your full peer review and any attached files.

Reviewer #1: No

Reviewer #2: No

---

## [Author Response · Author response to Decision Letter 0]

30 Nov 2020

We would like to thank the reviewers for a thorough and excessive job of reviewing this manuscript. The

inputs and suggestions from the referees have been taken under consideration and detailed responses to

reviewers are given in the attached file: Response to Reviewers

---

## [Decision Letter · Decision Letter 1]

23 Dec 2020

PONE-D-20-29309R1

Force-velocity profiling in athletes:

Reliability and agreement across methods.

PLOS ONE

Dear Dr. Lindberg,

Thank you for submitting your manuscript to PLOS ONE. After careful consideration, we feel that it has merit but does not fully meet PLOS ONE’s publication criteria as it currently stands. Therefore, we invite you to submit a revised version of the manuscript that addresses the points raised during the review process.

Please, address ASAP the revisions suggested by the reviewers to proceed with the acceptance of the manuscript.

We look forward to receiving your revised manuscript.

Kind regards,

Daniel Boullosa

Academic Editor

PLOS ONE

Reviewers' comments:

Reviewer's Responses to Questions

**Comments to the Author**

1. If the authors have adequately addressed your comments raised in a previous round of review and you feel that this manuscript is now acceptable for publication, you may indicate that here to bypass the “Comments to the Author” section, enter your conflict of interest statement in the “Confidential to Editor” section, and submit your "Accept" recommendation.

Reviewer #1: (No Response)

Reviewer #2: All comments have been addressed

2. Is the manuscript technically sound, and do the data support the conclusions?

Reviewer #1: Yes

Reviewer #2: Yes

3. Has the statistical analysis been performed appropriately and rigorously? 

Reviewer #1: Yes

Reviewer #2: Yes

4. Have the authors made all data underlying the findings in their manuscript fully available?

Reviewer #1: (No Response)

Reviewer #2: Yes

5. Is the manuscript presented in an intelligible fashion and written in standard English?

Reviewer #1: Yes

Reviewer #2: Yes

6. Review Comments to the Author

Reviewer #1: I would like to thank the authors for addressing most of my previous comments. I think the manuscript would benefit from revising some minor typos (please, see some examples below), but is overall acceptable for publication. It would be good if an English native speaker revises the writing (given that PLOS One does not copyedit accepted manuscripts).

Last line of abstract and conclusion: “as well as OF THE differences across measurement methods…” The of is missing.

Introduction. Line 63. Controversy “exists”. The “s” is missing.

Introduction. Line 67. Reference number 19 assessed between-session reliability, not within-session reliability.

Methods, Line 163-164: A protocol of…”was used”. Is the verb referring to “protocol” or to “the loads”?

Methods, Line 164: The increase in loads “was” then individually determined, or loads “were” then individually determined.

Methods, line 186-191: Please be consistent with the verb tense (past simple), i.e., "was gradually increased..."

Results, line 277: the typical error...was, or the typical errorS (in plural)...were

Discussion, Line 366 The correct citation would be Valenzuela et al. (Pedro L. is the name, not the surname). Also applicable in the reference list.

Discussion, Line 424: No need to mention the journal (PeerJ) here. Please, revise the citation.

Reviewer #2: Manuscript: PONE-D-20-29309R1

GENERAL COMMENTS: I would like to congratulate the authors for their effort to address the points raised. I believe that the quality of the paper has greatly improved to the level expected for its publication. The authors have reformulated their discussion which allows for a better understanding of the phenomenon being studied. This manuscript adds to the recent and compelling evidence questioning the reliability of the FV variables which has great implications for coaches and sport scientists. I am happy to endorse the publication of the manuscript. However, the authors should adjust the following minor details:

Tables 3 and 4. Tables 3 and 4. For consistency concerning the rest of the manuscript, please report the ICC values as "0.81" instead of ".81".

Line 366 (consider the line numbers of the manuscript version without marked changes). Please note, that the author is "Valenzuela et al. [19]" and not "Pedro L. et al." Also, this reference should be corrected in the reference list.

Line 424. The reference “(Helland et al 2020 PeerJ)” must be corrected so per journal style. In addition, I was not able to find this reference in the reference list.

Line 529. Please replace "This variations" with "These variations".

7. PLOS authors have the option to publish the peer review history of their article (what does this mean?). If published, this will include your full peer review and any attached files.

Reviewer #1: No

Reviewer #2: No

---

## [Author Response · Author response to Decision Letter 1]

7 Jan 2021

Thank you for the useful comments. Please se the attached revised manuscript and response letter

---

## [Editor Report · Decision Letter 2]

8 Jan 2021

Force-velocity profiling in athletes:

Reliability and agreement across methods.

PONE-D-20-29309R2

Dear Dr. Lindberg,

We’re pleased to inform you that your manuscript has been judged scientifically suitable for publication and will be formally accepted for publication once it meets all outstanding technical requirements.

Kind regards,

Daniel Boullosa

Academic Editor

PLOS ONE
---

## [Editor Report · Acceptance letter]

21 Jan 2021

PONE-D-20-29309R2 

Force-velocity profiling in athletes: *Reliability and agreement across methods*. 

Dear Dr. Lindberg:

I'm pleased to inform you that your manuscript has been deemed suitable for publication in PLOS ONE. Congratulations! Your manuscript is now with our production department. 

Kind regards, 

on behalf of

Dr. Daniel Boullosa 

Academic Editor

PLOS ONE